# Transcriptional activator TAp63 is upregulated in muscular atrophy during ALS and induces the pro-atrophic ubiquitin ligase *Trim63*

Yannick von Grabowiecki[1,2], Paula Abreu[1,2], Orphee Blanchard[1,2], Lavinia Palamiuc[2,3], Samir Benosman[3], Sophie Mériaux[2,3], Véronique Devignot[1,2], Isabelle Gross[1,2], Georg Mellitzer[1,2], José L Gonzalez de Aguilar[2,4], Christian Gaiddon[1,2]*

[1]UMR_S 1113, Molecular mechanisms of stress response and pathologies, Institut national de la santé et de la recherche médicale, Strasbourg, France; [2]Fédération de Recherche Translationnelle, Strasbourg University, Strasbourg, France; [3]Sanford Burnham Medical Research Institute, San Diego, United States; [4]Institut national de la santé et de la recherche médicale, Laboratoire SMN, Strasbourg, France

**Abstract** Mechanisms of muscle atrophy are complex and their understanding might help finding therapeutic solutions for pathologies such as amyotrophic lateral sclerosis (ALS). We meta-analyzed transcriptomic experiments of muscles of ALS patients and mouse models, uncovering a p53 deregulation as common denominator. We then characterized the induction of several p53 family members (p53, p63, p73) and a correlation between the levels of p53 family target genes and the severity of muscle atrophy in ALS patients and mice. In particular, we observed increased p63 protein levels in the fibers of atrophic muscles via denervation-dependent and -independent mechanisms. At a functional level, we demonstrated that TAp63 and p53 transactivate the promoter and increased the expression of *Trim63* (MuRF1), an effector of muscle atrophy. Altogether, these results suggest a novel function for p63 as a contributor to muscular atrophic processes *via* the regulation of multiple genes, including the muscle atrophy gene *Trim63*.

*For correspondence: gaiddon@unistra.fr

Competing interests: The authors declare that no competing interests exist.

## Introduction

Muscle atrophy is associated with aging, cancer, AIDS and neurodegenerative diseases such as amyotrophic lateral sclerosis (ALS) (*von Haehling et al., 2010*). Although muscle atrophy is not necessarily the primary target of the pathology, it is often an important cause of lethality. For example, atrophy and dysfunction of respiratory muscles lead to death in ALS patients (*Rothstein, 2009*). As muscle atrophy is associated with complex pathologies, the exact mechanisms inducing muscle atrophy are varied and still debated. Typically, ALS has been considered a neurodegenerative pathology specifically causing alteration in motor neurons, but more recent findings indicate that the etiology of the pathology is more complex. Indeed, a number of additional cell types, such as astrocytes (*Yamanaka et al., 2008*), microglia (*Boillée et al., 2006*) and muscle cells (*Wong and Martin, 2010*), have been described to be directly affected by the pathology and therefore to participate in the muscle atrophy.

Around 20% of all inherited ALS cases can be linked to mutations in the gene encoding SOD1. Cellular events (*Pansarasa et al., 2014*) that have been shown to be triggered by these different mutations include aggregation of SOD1 proteins in the cytoplasm (*Hart, 2006*), increase in oxidative

**eLife digest** Many conditions, including cancer and AIDS, lead to a person's muscles wasting away. Often the muscles are not necessarily the prime targets of these diseases. However, in the case of a neurodegenerative disease called amyotrophic laterals sclerosis (or ALS), it is the muscle degeneration that ultimately leads to the death of patients within a few years. There is currently no treatment for ALS. This is partly because the mechanisms behind the disease are poorly understood. However, proteins belonging to the so-called p53 family have been implicated as possibly being involved in the degenerative processes in muscles.

The p53 family comprises three proteins called p53, p63 and p73. These proteins bind to DNA to switch genes on or off, and target genes involved in a range of cellular processes such as DNA repair and cell death. von Grabowiecki et al. have now identified p63 as a protein that contributes to muscle wasting in ALS. The production of p63 in degenerated muscles increases as the disease progresses, both in patients and in a mouse model of the condition. Further analysis then showed that the protein p63 regulates an enzyme that actually breaks down the building blocks of muscles, which directly leads to the muscle wasting.

Following on from this work, the next step is to verify if the p53 family are also involved in muscle wasting that arises from other diseases. This will enable researchers to look for similarities that could highlight new opportunities to treat conditions that cause muscle wasting.

stress (*Barber and Shaw, 2010*) and subsequent DNA damage (*Aguirre et al., 2005*), endoplasmic reticulum (ER) stress (*Nishitoh et al., 2008*) or alterations of mitochondrial function (*Manfredi and Xu, 2005*). In addition, novel mutated genes (*FUS, TARDBP...*) have been linked to ALS with differences in the pathophysiological outcomes (*Chen et al., 2013*). These differences might be linked to the different impacts of the mutated proteins at the molecular level. Indeed, protein aggregates or other alterations induced by SOD1 mutants have been characterized in muscle cells, while other mutated proteins linked to ALS seem to not directly affect muscles (*Pansarasa et al., 2014*).

To date, the exact molecular mechanisms driving muscle catabolism in the symptomatic phase of ALS remain poorly understood. It is also only during the symptomatic phase that ALS pathology can be diagnosed. The absence of pre-symptomatic markers highlights the need for understanding the muscle catabolic processes for therapeutic purposes. Several observations indicate that the p53 family members (p53, p63, p73) play an important role in muscle physiopathology and might therefore represent actors of the muscle atrophy (*Schwarzkopf et al., 2006*; *Mazzaro et al., 1999*; *Cam et al., 2006*; *Fontemaggi et al., 2001*; *Martin et al., 2011*; *Rouleau et al., 2011*; *Belloni et al., 2006*; *Su et al., 2012*).

The p53 family of transcription factors is a central regulator of cellular processes such as apoptosis, cell cycle arrest, metabolism or cellular differentiation through the regulation of several target genes (*CDKN1A, BAX, GADD45A, MDM2* and others) (*Arrowsmith, 1999*; *Menendez et al., 2009*). All three members encode TA and △N isoforms that vary in their N-terminus due to alternate promoter usage where TA has a canonical transactivation domain. △N isoforms lack such a domain and can serve as dominant negatives versus the TA isoforms in some cases, although they are also capable of transactivating certain genes (*De Laurenzi et al., 1998*; *Casciano et al., 1999*; *Murray-Zmijewski et al., 2006*).

Through their cellular activities, p53 proteins are involved in a broad variety of physiological functions that include tumor suppression and organ development (*Arrowsmith, 1999*). For example, p53 plays a role in the response against tumor-inducing events such as DNA damage, oncogene activation, and a variety of additional cellular stresses (hypoxia, reactive oxygen species (ROS), or alteration of energy metabolism) (*Marcel et al., 2011*; *Rufini et al., 2013*; *Gonfloni et al., 2014*). In addition, several studies have highlighted the involvement of the p53 family members in neurodegenerative diseases. p53 as well as p63 and p73 have been shown to regulate neuronal apoptosis and their activation has been observed in various neurodegenerative diseases, such as Alzheimer, Parkinson and Angelman syndromes (*Jiang et al., 1998*; *de la Monte et al., 1997*; *Seidl et al., 1999*; *Bui et al., 2009*; *Benosman et al., 2007*; *Benosman et al., 2011*). We have previously

reported an induction of p53 in degenerating spinal cord motor neurons in an ALS mouse model expressing mutated Cu/Zn superoxide dismutase 1 (SOD1[G86R]) (*González de Aguilar et al., 2000*).

In muscles, p53 is activated during myogenic differentiation, participates with MyoD to induce myogenesis, and mediates doxorubicin-induced muscle atrophy via its target gene *pw1* (*Schwarzkopf et al., 2006*; *Mazzaro et al., 1999*). Nonetheless, p53 expression is not essential for muscle development (*Donehower et al., 1992*) or regeneration (*White et al., 2002*), which could be explained by compensatory mechanisms involving p63 and p73. Indeed, more recent studies have shown that p63 and p73 are also involved in myoblast differentiation (*Cam et al., 2006*; *Fontemaggi et al., 2001*; *Martin et al., 2011*; *Rouleau et al., 2011*) and ΔNp73 appears to protect muscle cells against stresses (*Belloni et al., 2006*). Finally, a study showed that p63 is important for the regulation of muscle cell metabolism via the regulation of Sirtuins and AMPK (*Su et al., 2012*).

In this study, we investigated the regulation and the role of the transcription factors of the p53 family in muscular atrophy during ALS based on a meta-analysis we performed with 4 microarray experiments obtained with biopsies of muscles from ALS patients or with muscles from ALS mouse models.

## Results

### p53-target genes and p53 regulators are induced in atrophic muscles during ALS

To identify the molecular mechanisms involved in muscle atrophy during ALS we performed a meta-analysis using four independent microarray experiments deposited at the Array Express database (EMBL-EBI). Two experiments contained gene expression data for the muscle of ALS patients and control individuals (E-MEXP-3260; E-GEOD-41414, [*Pradat et al., 2012*; *Bernardini et al., 2013*]). One experiment contained gene expression data for muscles of SOD1(G86R) mice that represents an ALS model in which the onset of the pathology is at 105 days of age (E-TABM-195 [*Gonzalez de Aguilar et al., 2008*]). The last experiment contained gene expression data for muscles of SOD1 (G93A) mice in which onset of the pathology occurs at 14 weeks of age (E-GEOD-16361, [*Capitanio et al., 2012*]). Beside the better pathophysiological relevance, data obtained from biopsies of ALS patients also provided a better representation of the diversity of the genetic anomalies observed in patients. In addition, patients were at various stage of the pathology, hence establishing a representative scale of muscle alterations. The panel of datasets we chose also included two different mouse models of ALS, allowing us to pinpoint common and specific deregulations. Importantly, the SOD1 mouse models are well characterized for their muscular phenotype alterations. In particular, it has already been established that SOD1 mutants present altered functions in muscles, in contrast to other mutated proteins linked to ALS (TARDBP, FUS etc) (*Pansarasa et al., 2014*).

After standard normalization and statistical analyses, each experiment was independently subjected to gene ontology, signaling pathway, transcription factor, and miRNA analyses. Fold induction between control individuals and ALS individuals was set to twofold change and rawp value inferior to 0.05. We decided to focus on transcription factor deregulations. The bioinformatic analyses we performed pinpointed to only 7 transcription factors whose activity, indicated by coherent changes in expression of their target genes, was potentially deregulated in at least two out of four experiments (*Figure 1A*). The activity of one transcription factor, NfKB, appeared deregulated only in experiments done with the mouse models. Deregulation of STAT1 activity was identified in three experiments. Interestingly, the activity of only three transcription factors, MyoD, Myogenin and p53, was identified to be commonly deregulated in all four experiments that included biopsies from patients and animal models. MyoD and Myogenin are muscle specific transcription factors involved in muscle cell differentiation (*Zanou and Gailly, 2013*). P53 was the transcription factor with the highest number of deregulated genes (51 genes). Notably the p53 target genes *CDKN1A*, *GADD45A* and *PMAIP1*, among others, were found induced in all four experiments.

As one of the experiments using biopsies of ALS patients included a scale (*von Haehling et al., 2010*; *Rothstein, 2009*; *Yamanaka et al., 2008*; *Boillée et al., 2006*; *Wong and Martin, 2010*; *Pansarasa et al., 2014*) of muscle alteration, we analyzed whether the expression of some of these genes might correlate with the severity of the pathology. We found that *CDKNA1*, *GADD45A* and

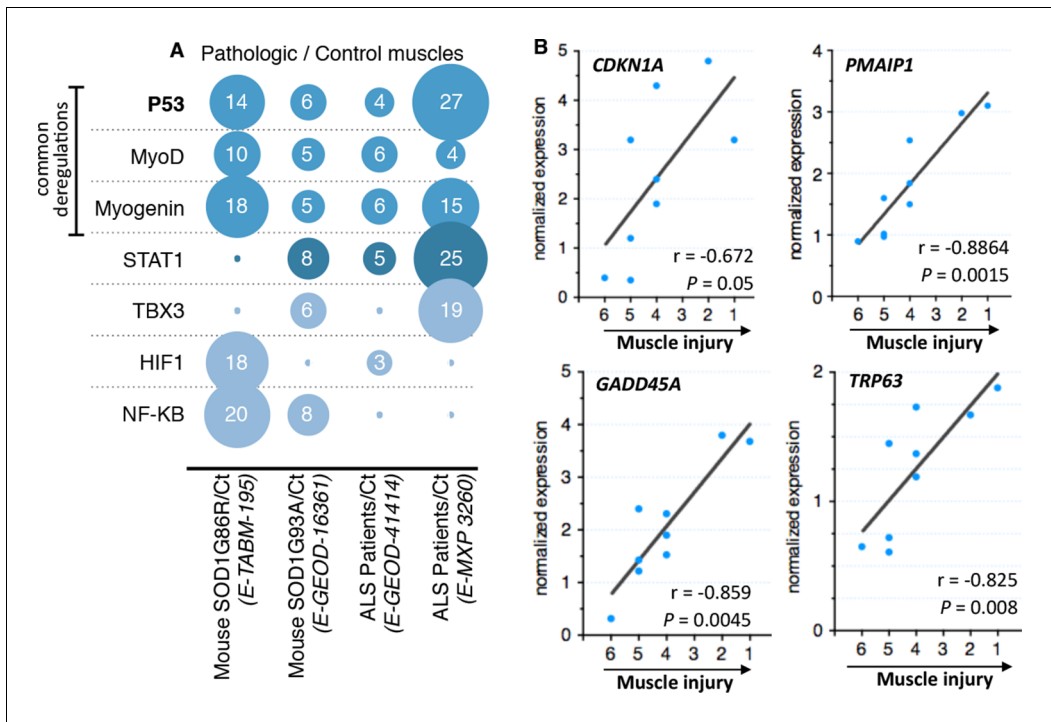

**Figure 1.** Microarray meta-analysis highlights links between the deregulation of p53 family related genes and ALS. (A) Representation of the number of deregulated target genes of the indicated transcription factors. Data were obtained using the indicated datasets from the Array Express database (EMBL-EBI) and quantification was carried out from AltAnalyze software analyses on transcription factor databanks (complete data in *Supplementary file 2A, B,C,D*. (B) mRNA levels from nine ALS patient deltoid muscles as by DNA microarray were correlated with the intensity of muscle injury. Expression data were generated using a murine gene profiling database deposited at ebi.ac.uk/arrayexpress (accession number E-MEXP-3260). In the corresponding study, muscle injury was estimated according to a composite score combining manual testing of strength of shoulder abductors and the degree of myofiber atrophy. This score ranges from 6 (normal strength and very low level of atrophy) to 1 (total paralysis and high level of atrophy). Each point represents an individual. Correlation coefficients (r) and p-values were determined by Spearman correlation test.

The following figure supplements are available for figure 1:

**Figure supplement 1.** Regulation of p53-family related genes in skeletal muscle of SOD1(G86R) and denervated mice.

**Figure supplement 2.** mRNA levels from control and ALS patient deltoid muscles as by DNA microarray were correlated with the intensity of muscle injury.

---

*PMAIP1* expression correlated with the degree of the pathology of the muscle from ALS patients (*Figure 1B*).

Besides the bioinformatic analysis on the deregulation of transcription factors, the signaling pathway analyses also indicated alterations in the p53 pathway characterized by deregulations in upstream regulators of p53, such as *MDM2* and thioredoxin, and a p53 family member, *P63* (*Table 1*, *Figure 1B*, *Figure 1—figure supplement 1*). In particular, the expression of *P63* correlated with the severity of the pathology in muscles biopsies from ALS patients (*Figure 1B*).

In order to validate the bioinformatic analyses we performed RT-qPCR experiments with RNA from muscle biopsies of an independent group of ALS patients. We confirmed that *CDKN1A*, *GADD45A* and *PMAIP1* were induced in the muscle biopsies of ALS patients (*Figure 2A,B,C*). Similarly, we analyzed the expression of these genes using muscle samples of independent groups of SOD1(G86R) mice. Groups analyzed at 60 days and 75 days of age correspond to the asymptomatic stage, while 90 day-old groups correspond to an early or pre-symptomatic stage associated with

**Table 1.** Fold induction of p53-related genes in the ALS model SOD1 (G86R).

| Gene name | Function | 90 d. | 105 d. |
|---|---|---|---|
| **p53-family target genes** | | | |
| Cdkn1a (p21) | Cell cycle arrest | 4 | 13 |
| Gadd45a | Cell cycle arrest | 5,6 | 21 |
| Peg3 | Apoptosis inducing | 3 | 7 |
| Perp | Cell cycle arrest | 4 | 12 |
| Pmaip1 | Apoptosis effector | 5 | 12 |
| Bax | Apoptosis effector | 3 | 8 |
| Siva | Apoptosis inducing | 3 | 5 |
| Zmat3 | Growth regulation | 1,6 | 1,1 |
| Eda2R | NF.Kb/JNK pathway | 3,4 | 9,4 |
| Tigar | Glucose metabolism | 0,75 | 0,2 |
| Sens1 | ROS homeostasis | - | 16,3 |
| Sens2 | ROS homeostasis | 1,27 | 1,46 |
| Sco2 | Glucose metabolism | 1,18 | 0,91 |
| Ddit3 (Chop) | ER stress | 1,14 | 0,35 |
| Bip (Grp78) | ER stress | 1,25 | 1,08 |
| Xbp1 | ER stress | 2 | 2,51 |
| **p53-family regulators** | | | |
| Mlf1 | Cell cycle arrest/differentiation | 0,9 | 0,2 |
| Myf6 | Differentiation | 4 | 7 |
| Mdm2 | p53 degradation | 4 | 6 |
| Txn1 | Oxidative stress response | 4 | 6 |
| Id2 | Inhibition of differentiation | 2 | 3,1 |
| **p53-family members** | | | |
| P53 | | 4 | 3 |
| TAp63 | | 4 | 12 |
| ΔNp63 | | 0,5 | 0,3 |
| TAp73 | | 2 | 3 |
| ΔNp73 | | 0,9 | 0,8 |
| **Denervation/atrophy markers** | | | |
| Chrna1 (ACh Receptor alpha) | Neuromuscular junction | 4,2 | 12,4 |

established gene deregulations (*von Grabowiecki et al., 2015*). Finally, the symptomatic stage group (beginning after 105 days) is characterized by the onset of paralysis and marked muscle atrophy (*Figure 2—figure supplement 1* Upregulation of the p53 target genes *Gadd45a, Cdkn1a, Bax, Pmaip1 and Perp* was observed at 90 days and further increased at 105 days in SOD1(G86R) mice (*Figure 2D,E,F*, Figure 2—figure supplement 2). In addition to these genes, we also analyzed by RT-qPCR the expression of additional p53 target genes and regulators of p53 by RT-qPCR (*Table 1*). In particular, the expression of p53 target genes involved in apoptosis (*Pmaip1, Peg3* and *Siva*) was also induced.

p53 proteins have recently been linked to energy metabolism and endoplasmic reticulum (ER) stress pathway activation (*Su et al., 2012*; *Ramadan et al., 2005*; *Zhu and Prives, 2009*). Analysis of the expression of p53 family target genes implicated in several metabolic pathways (*Tigar*, sestrins, *Sco2*, Sirtuin1 or Prkaa1) (*Su et al., 2012*; *Vousden and Ryan, 2009*) or ER stress (*Chop, Bip*, or *Xbp1*) (*Stavridi and Halazonetis, 2004*), did not reveal coherent regulation in respect to disease progression (*Table 1*). For example, the expressions of *Sesn2* and *Tigar* (*Vousden and Ryan, 2009*)

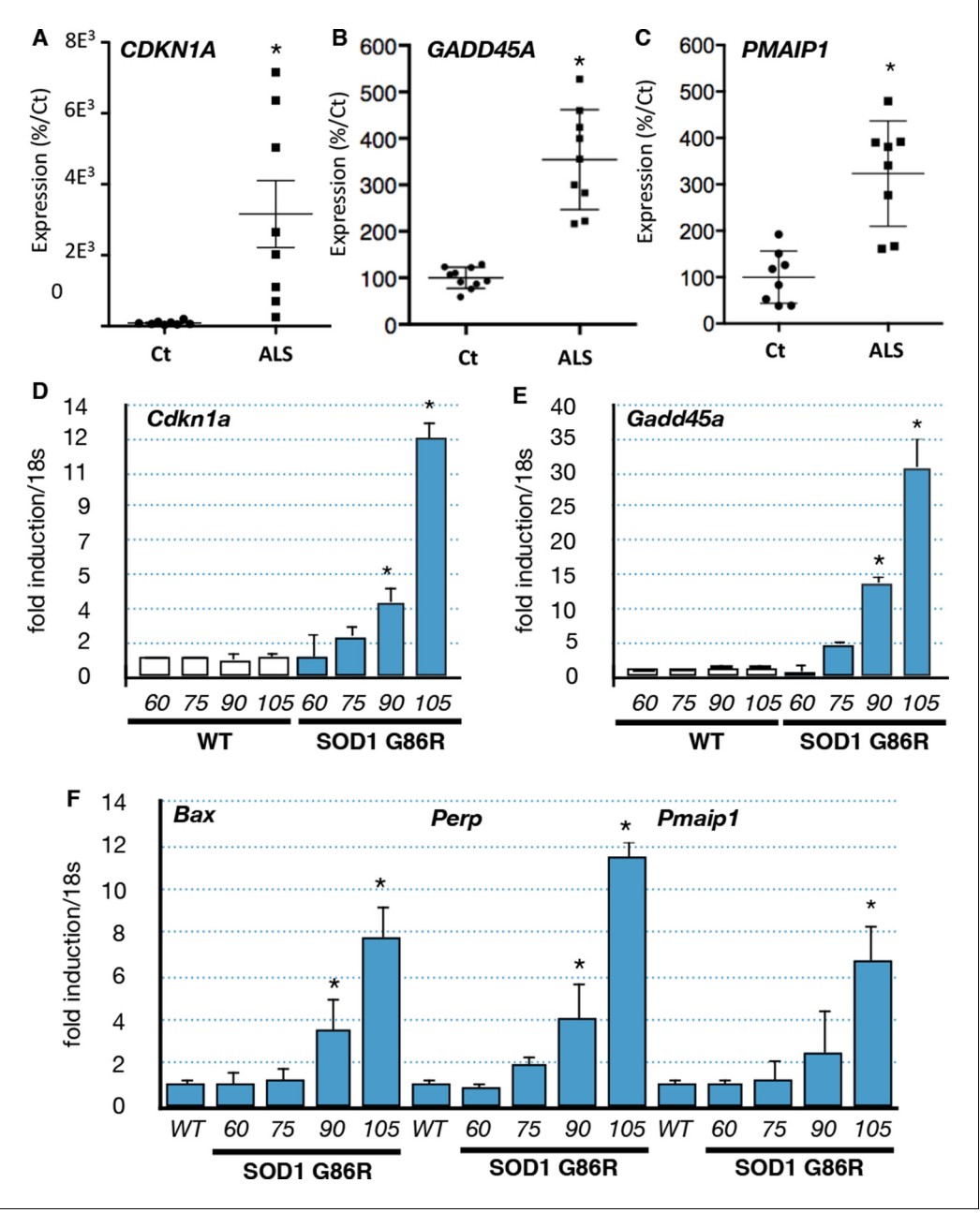

**Figure 2.** p53-family target gene expression in muscles from ALS patients and in an ALS mouse model correlates with disease intensity. (**A–C**) RNA from muscle biopsies of control and ALS patients (n = 8, Neuromuscular Unit [BioBank of Skeletal Muscle, Nerve Tissue, DNA and cell lines]) was extracted and analyzed by RT-qPCR. Absolute levels are normalized against the average of the control group. (**D–F**) p53 family target genes mRNA levels were assayed in SOD1(G86R) mouse gastrocnemius muscle by RT-qPCR. Graphs are means of fold induction versus 60 days-old WT and of matching age (60, 75, 90, 105-days-old, n = 6) and experimental condition (wild-type, or SOD1 (G86R)). *p<0.01 compared to control, as calculated by a one-way ANOVA test followed by a Tukey post-test.

The following figure supplement is available for figure 2:

**Figure supplement 1.** Gastrocnemius muscles from wild-type or symptomatic SOD1(G86R) (105 days) mice were dissected and weighted.

were regulated in opposite directions during the progression of the disease. Therefore, our data suggest that the correlation between ALS progression and p53 function might mostly be due to cell growth arrest and cell death regulation.

We also confirmed by RT-qPCR an upregulation of several upstream regulators of the p53 family, including *Mdm2*, *Myf6*, *Mlf1*, and *Txn* (*Table 1*) (*Arrowsmith, 1999*). Taken together, our results suggest that a p53-like pathway is activated in ALS muscles both in patients and the murine SOD1 ALS-models.

## p53-family members are regulated in mouse skeletal muscles during ALS

As we observed in the muscle biopsies of ALS patients a correlation between p63 expression and the severity of the pathology, we investigated the expression levels of p53 family members in the muscles of SOD1(G86R) mice. Our analysis revealed an increased expression of TA isoforms of *Trp63* in SOD1(G86R) (*Figure 3*). Strikingly, the mRNA levels of TA isoforms of *Trp63* were strongly induced towards the end of the disease (105 day), while the mRNA levels for ΔN isoforms of *Trp63* were downregulated during the same time period. A similar tendency was observed for p53, TA and ΔN isoforms of *p73*, albeit at a lower magnitude. The expression of TA isoforms of *Trp63* correlated with acetylcholine receptor alpha (*Chrna1*) expression, a molecular marker indicating the severity of muscular denervation. In addition, we analyzed the expression of two documented effectors of muscular atrophy, namely *Fbxo32* (Atrogin-1) and *Trim63* (MuRF1). These proteins are E3 ubiquitin ligases that target muscular proteins for degradation during muscular atrophy or remodeling (*Murton et al., 2008*). Importantly, the deregulation of *Trp63* expression also correlated with the upregulation of these two markers. This is in accordance with our data from ALS patient muscle biopsies, whereby the expression of *P63* also correlated with the degree of muscle pathology (*Figure 1E*).

## p63 protein accumulates in muscle fibers during ALS

Based on the observed deregulation of *P63* expression in ALS patients and the stronger upregulation of TAp63 in SOD1(G86R) mice, we further analyzed p63 protein levels. Immunoblotting with a TAp63 isoforms specific antibody revealed a striking accumulation of p63 proteins in muscles of SOD1 (G86R) mice that correlated with the progression of the disease (*Figure 4A*). When probing with a ΔNp63 specific antibody, however, we did not observe any specific band. The use of a p63 antibody directed against all p63 isoforms confirmed an upregulation of p63 in muscles of SOD1 (G86R) mice (*Figure 4—figure supplement 1*). Immunohistochemistry with the same antibody also revealed markedly increased immunoreactivity in the nuclei of muscle fibers of SOD1(G86R) (*Figure 4B*, *Figure 4—figure supplement 3*). In contrast, there was no significant increase in p73 staining (*Figure 4—figure supplement 2*). In this case, the apparent higher number of p73 positive nuclei appeared to be due to the atrophy of the muscle fibers, increasing the density of cells/nuclei. Similar experiments to detect expression of p53 did not yield a specific staining. However, we observed by western blot some slight increase in p53 protein levels in protein extract of muscle from SOD1(G86R) mice (*Figure 4—figure supplement 1*). Taken together, our data indicated a complex regulation of p53 family members during muscular atrophy, highlighted by significant increase of TAp63 messenger and protein expression levels in the skeletal muscles during ALS.

## Muscle denervation induces a p63 response

We then further investigated the possible cause of the deregulation of *Trp63* expression. Several studies showed that the ALS etiology is complex and multifactorial, involving different cell types and molecular mechanisms. One established cause of muscular atrophy is motor neuron degeneration that leads to muscle denervation. However, it has also been shown that SOD1 mutants can also directly cause alteration in muscle cells such as SOD1 protein aggregates and mitochondrial abnormalities (*Pansarasa et al., 2014*). To verify the first hypothesis, we induced denervation in 80 day-old wild type and SOD1(G86R) mice by sciatic nerve crush, and gastrocnemius muscles were analyzed 7 days later. Our results showed that denervation upregulated TAp63 mRNA levels five- to sixfold in wild-type mice (*Figure 5A*). Concomitantly, ΔNp63 levels were downregulated 0.4-fold (*Figure 5B*). In SOD1(G86R) mice, nerve crush further accentuated changes in mRNA levels for TA

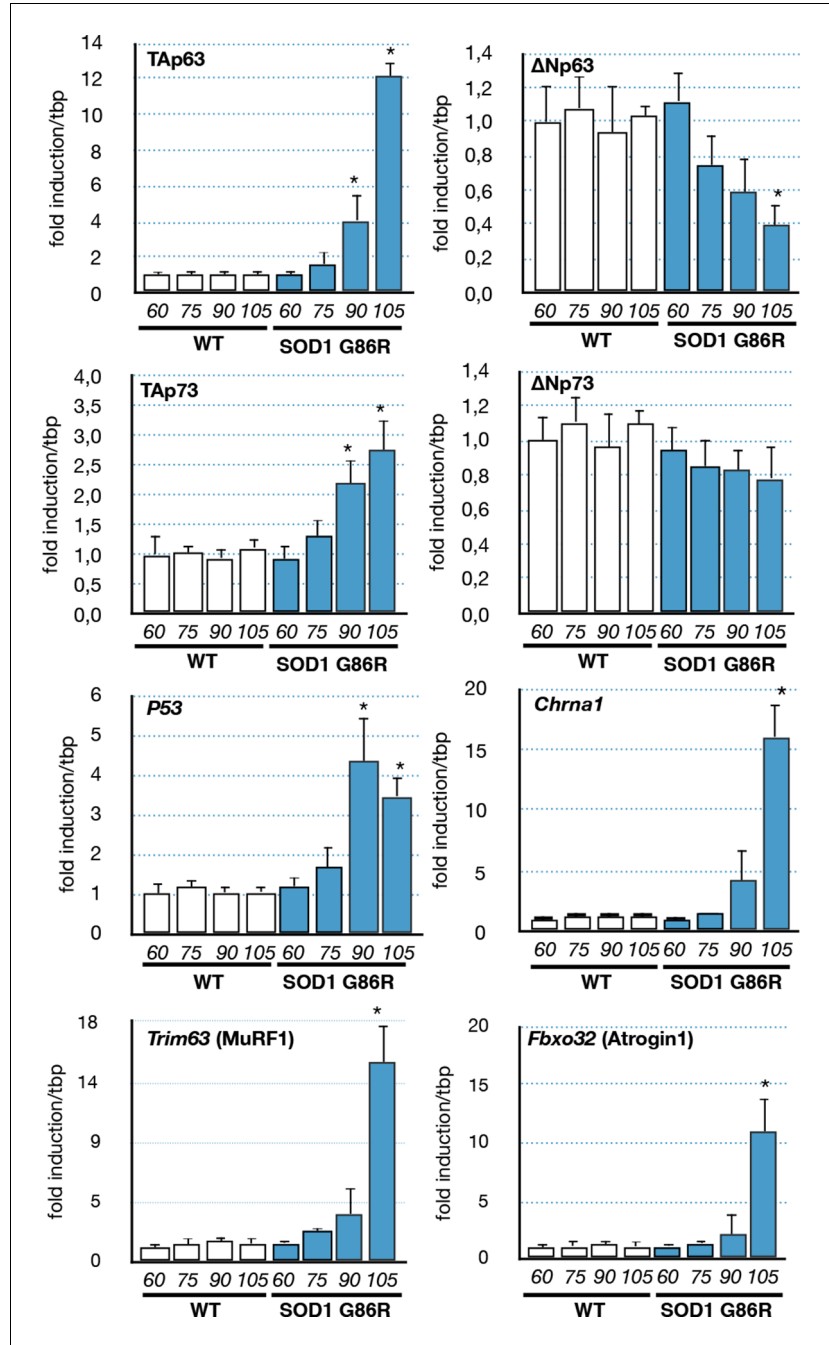

**Figure 3.** Expression of p53-family members in SOD1(G86R) muscles. p53 family members, *Chrna1* (Acetylcholine receptor subunit alpha) or muscle atrophy effectors *Trim63* (MuRF1) and *Fbxo32* (Atrogin1) mRNA levels were assayed in SOD1(G86R) mouse gastrocnemius muscle by RT-qPCR. Bars are means of fold induction versus 'WT 60 days-old' and of matching age (60, 75, 90, 105 days-old, n = 6) and experimental condition (WT or SOD1(G86R)). *p<0.01 compared to control, as calculated by a one-way ANOVA test followed by a Tukey post-test.

and ΔN isoforms of *Trp63*. In addition, the TAp63 target genes *Cdkn1a* and *Gadd45a* were found strongly induced after nerve crush (*Figure 5C,D*). These results show that nerve injury leading to alteration of the motor axis seems to be sufficient to activate a TAp63 response.

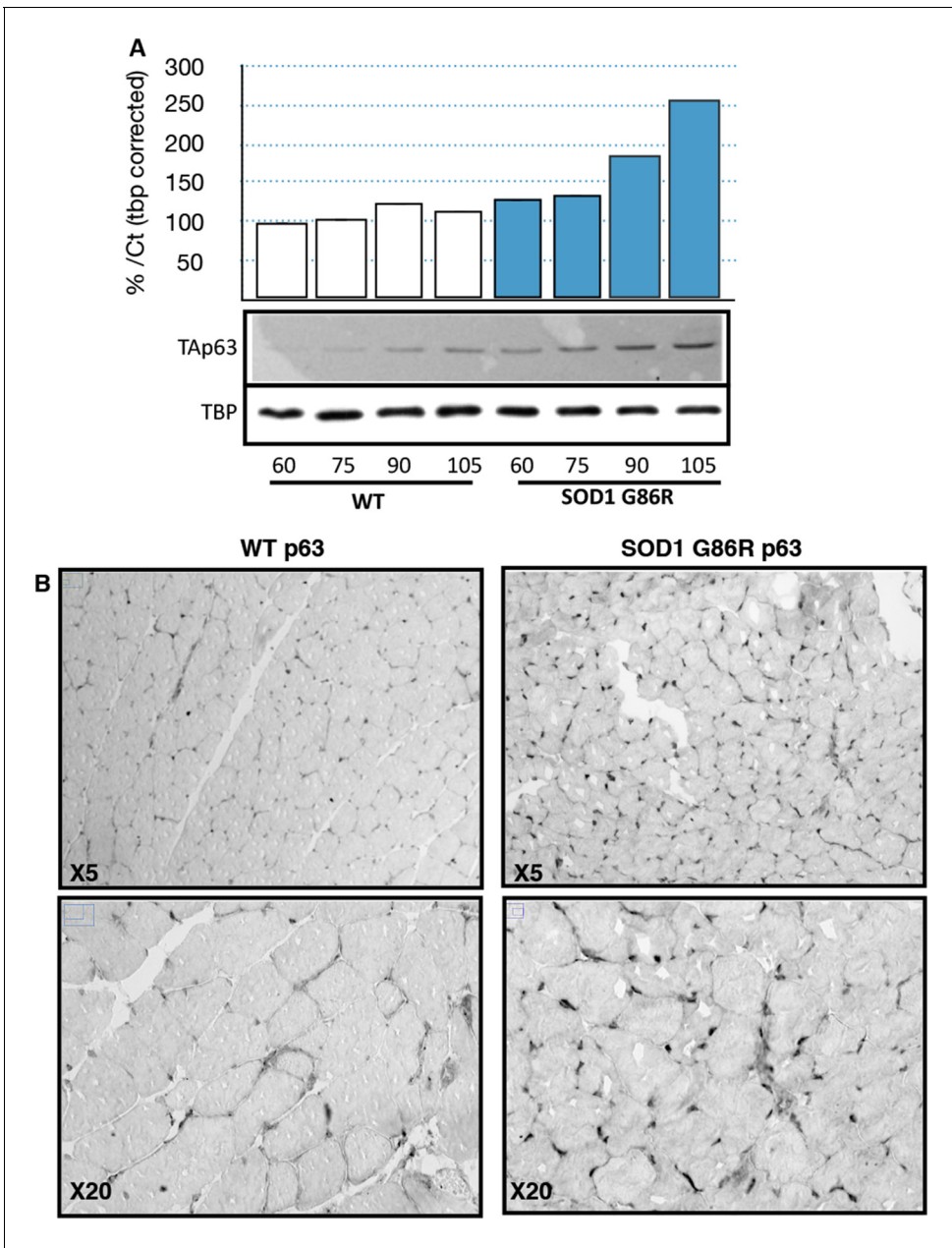

**Figure 4.** p63 protein expression in SOD1(G86R) muscle. (**A**) Proteins from muscles were immuno-precipitated with a p63 antibody and then separated on a 10% SDS PAGE gel. Western blot experiment was performed using an antibody against TAp63. Each experimental point is a pool of proteins from 6 animals. Graph represents quantification of the blot using ImageJ image analyzer software indicated a %/WT 60 day-old animals. (**B**) Gastrocnemius muscles from wild-type or symptomatic SOD1(G86R) (105 days) mice were cryodissected and probed for total p63 protein.

The following figure supplements are available for figure 4:

**Figure supplement 1.** p53 and p63 protein expression in muscles of SOD1(G86R) mice.

**Figure supplement 2.** Gastrocnemius muscles from wild-type or symptomatic SOD1(G86R) (105 days) mice were cryodissected and probed for total p73 protein.

**Figure supplement 3.** Gastrocnemius muscles from wild-type or symptomatic SOD1(G86R) (105 days) mice were cryodissected and probed for total p63 protein and nuclei (Hoechst).

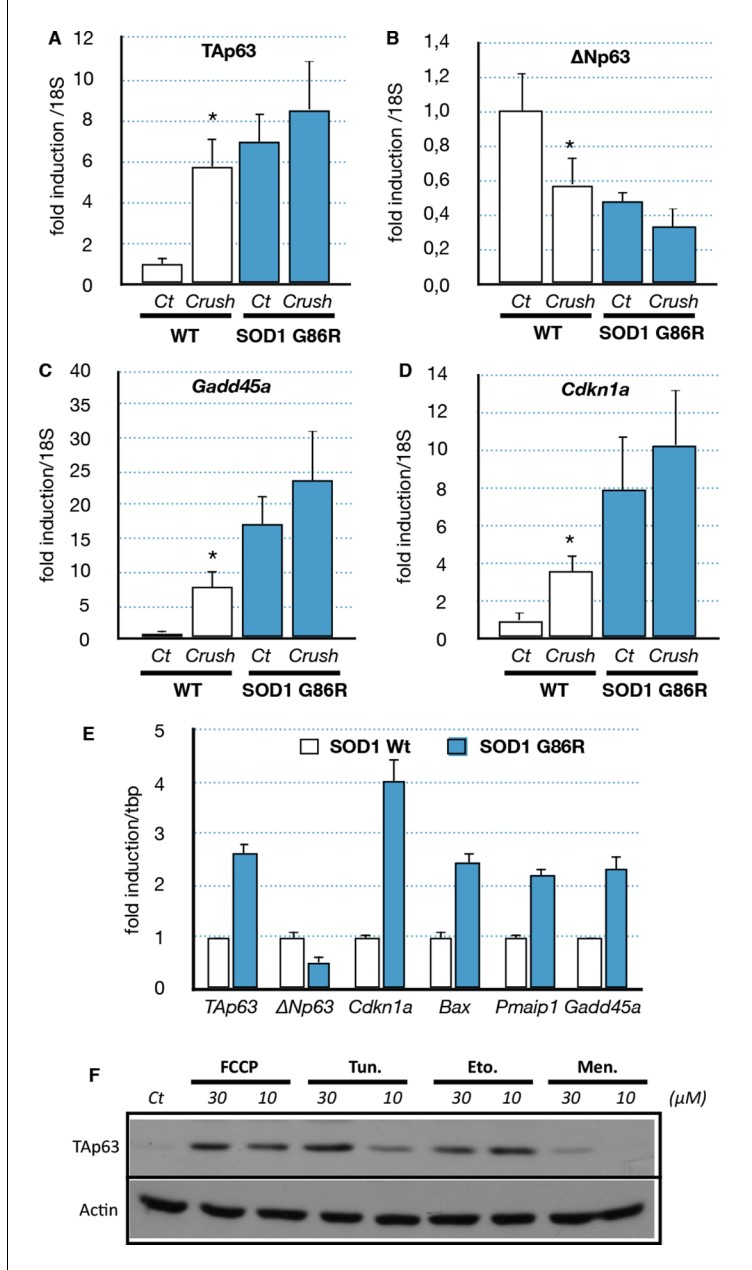

**Figure 5.** Expression of p63 and p53-family target genes following sciatic nerve crush, SOD1 expression of induction of stress (**A–D**) WT and SOD1 (G86R) mice (80 days of age) were anesthetized and the sciatic nerve crushed. Sham-operated contra limbs served as control (Ct). After 7 days, expression of TA isoforms of *Trp63* (**A**, TAp63), ΔN isoforms of *Trp63* (**B**, ΔNp63), *Gadd45a* (**C**) and *Cdkn1a* (**D**) was assayed by RT-qPCR (n = 6). Values were normalized to the value of sham-operated WT muscles/animals. Bars represent means (relative induction versus Ct) with standard deviation (n = 3). *p<0.01 as calculated by a one-way ANOVA test followed by a Tukey post-test. (**E**) C2C12 myoblasts were transfected with expression vectors for SOD1 variants (WT or G86R). mRNA from SOD1 transfected cells were analyzed by RT-qPCR for p63 and p63 target gene expression. Bars represent means (relative induction versus Ct) with standard deviation (n = 3). *p<0.01 as calculated by a one-way ANOVA test followed by a Tukey post-test. (**F**) Proteins were extracted from C2C12 myoblasts treated with compounds: FCCP, Tunicamycin (Tun), Etoposide (Eto), menadione (Men). Western blot analysis revealed TAp63 expression.

The following figure supplements are available for figure 5:

**Figure supplement 1.** Regulation of *p63* and *Mdm2* expression by SOD1 (G86R).

**Figure supplement 2.** Functional interaction between members of the p53 family and ER or mitochondrial stress.

## Mutated SOD1 is sufficient to induce the p63 response in myoblasts

Although it remains a challenge to reproduce in vitro the long-term development of ALS, we tried to assess the effect of the mutated SOD1 on muscle cells via an overexpression of SOD1(G86R) in the mouse myoblast cell line C2C12. Several target genes of the p53-family (*Bax, Cdkn1a, Gadd45a*) were induced upon overexpression of SOD1(G86R) (*Figure 5E*). Similarly TAp63 expression was increased at the mRNA level and the protein level (*Figure 5E* and *Figure 5—figure supplement 1*). In contrast, the mRNA levels as well as the promoter activity of △N isoforms of *P63* were downregulated (*Figure 5E*, *Figure 5—figure supplement 1*) (*Romano et al., 2006*). However, we were not able to confirm this result on △Np63 at the protein level. We also tested for a possible cross-regulation of △Np63 expression by the increased expression of TAp63 observed in ALS. We observed that TAp63 represses the promoter activity of the △N isoforms of *P63*, while it expectedly induces the *Mdm2* promoter (*Figure 5—figure supplement 1*). These results demonstrated that expression of SOD1(G86R) was sufficient to trigger a p53-like response similar to our in vivo observations in atrophic muscle tissues.

We then investigated whether TAp63 could be induced by different stresses related to the cellular damages caused by SOD1 mutants. We used pharmacological inductors for oxidative stress (menadione) (*Barber and Shaw, 2010*), DNA damage (etoposide) (*Aguirre et al., 2005*), mitochondrial deregulation (FCCP) (*Manfredi and Xu, 2005*) and ER Stress (tunicamycin) (*Hart, 2006*; *Nishitoh et al., 2008*). Treated cells revealed an increase of TAp63 upon the four stresses (*Figure 5F*, *Figure 5—figure supplement 2*). Mitochondrial and ER stress triggers specific signaling pathways that involve a complex network of transcription factors such as ATF4, ATF6, XBP1 and CHOP (*Senft and Ronai, 2015*). Interestingly, overexpression of ATF4 and ATF6 induces the RNA level for TAp63 and TAp73 respectively, but not p53 (*Figure 5—figure supplement 2*). This result indicated that upregulation of TAp63 expression might be involved in the muscle cell response to diverse stresses including stresses related to SOD1 mutants.

## TAp63 regulates *Trim63* (MuRF1) expression, a specific muscle atrophy effector

As the expression profile of the TA isoforms of *Trp63* correlated with the expression of the muscle atrophy effectors *Fbxo32* and *Trim63* (*Figure 3*), we hypothesized that TAp63 could regulate them directly. Bioinformatic analyses revealed the presence of several putative p63-binding sites in the promoter of *Trim63* (*Figure 6A*). Therefore, we tested whether TAp63 could regulate *Trim63* expression. Indeed, TAp63 overexpression in C2C12 cells strongly induced *Trim63* mRNA levels (*Figure 6B*). *Fbxo32* expression level was much less affected (data not shown). Note that under this condition p53 or p73 had less effect on *Trim63* expression (*Figure 6B*). Under the same experimental conditions, other p63 target genes, *Cdkn1a, Pmaip1, Casp1* and *Prkaa1* were less induced (*Figure 6C*).

To further characterize the regulation of *Trim63* by TAp63, we used luciferease reporter constructs containing progressive deletions of the *Trim63* promoter. We found that p53 family members induced *Trim63* promoter reporters that contained at least the fragment -500 bp to -1000 bp (*Figure 7A and B*) (*Waddell et al., 2008*). Interestingly, that fragment contains potential p63 binding sites with high probability scores, such as RE1/2 (-660/-690 bp). We then assessed the capacity of p63 to bind the *Trim63* promoter on binding sites that have high probability scores. Chromatin immunoprecipitation experiments (ChIP) covering RE1/2 and RE4 binding sites showed that TAp63 proteins bound preferentially onto RE1/2 (*Figure 7C*). Similarly, ChIP experiments indicated that p73 and p53 bound to RE1/2 (*Figure 7—figure supplement 1*). However, p73 seemed also to bind RE4.

To assess the physiological importance of TAp63 in *Trim63* expression we used TAp63-specific silencing RNA (siRNA). Transfection in C2C12 cells of TAp63siRNA diminished the expression of TAp63 at the protein and mRNA levels (*Figure 7—figure supplement 1*). TAp63 silencing or overexpression of △Np63 had a partial protective effect on C2C12 (*Figure 7—figure supplement 2*). Importantly, silencing of TAp63 reduced *Trim63* mRNA levels in both basal state and following stress induced by FCCP (*Figure 7D*). SiRNA against p53 also diminished *Trim63* RNA level, while siRNA against p73 had not significant effect (*Figure 7—figure supplement 2*). The combination of siRNA against TA isoforms of *Trp63*, TA isoforms of *P73* and *P53* diminished further *Trim63* RNA level up to ~50%, but did not abolish it. Taken together, these results indicate a complex regulation of the

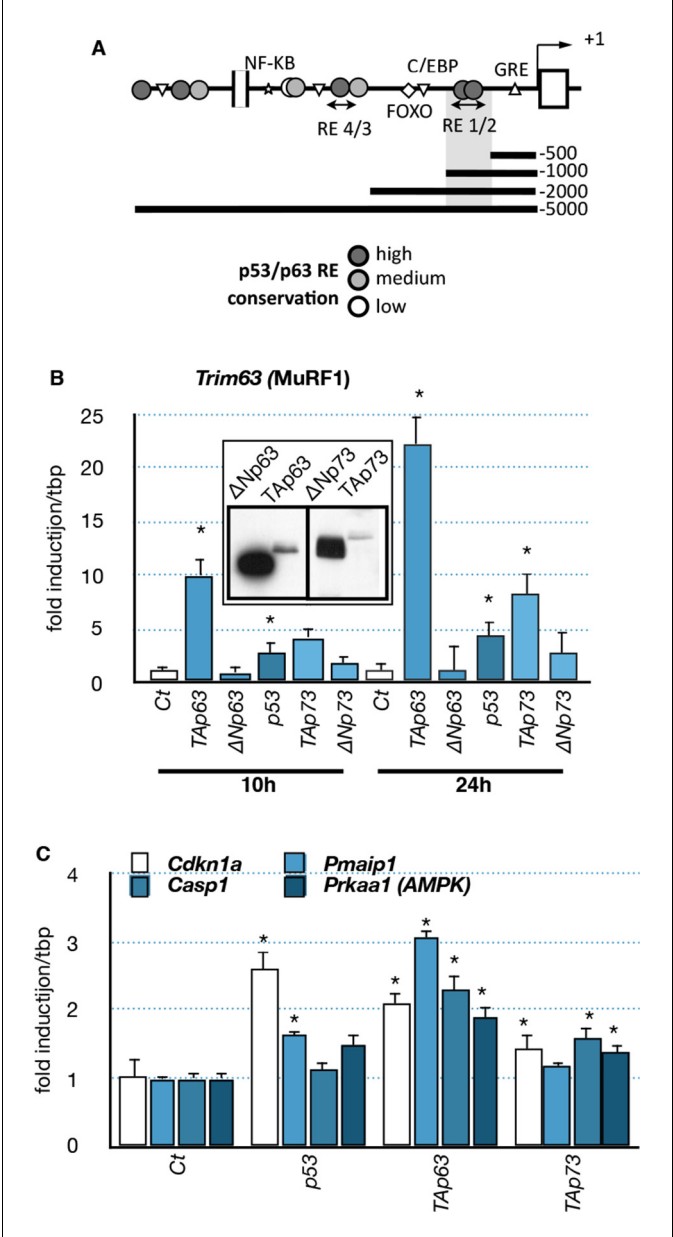

**Figure 6.** Effects of p53-family expression on *Trim63* and p53-family target genes. (**A**) Schematic representation of the *Trim63* promoter indicating the location of putative p53/p63 binding sites. (**B, C**) C2C12 myoblasts were transfected (inserted panel: western blot) with various p53-family members (TAp63γ, ΔNp63γ, p53, TAp73β, ΔNp73β). Total C2C12 RNA was subjected to RT-qPCR after 10 hr or 24 hr of transfection and *Trim63* (**B**) or p63 target (**C**, *Cdkn1a, Pmaip1, Casp1, Prkaa1*) expressions are shown relative to control-transfected cells. Bars are means of fold induction versus the control (Ct) with SD (n=3). *p<0.01 as calculated by a one-way ANOVA test followed by a Tukey post-test.

*Trim63* promoter, in which the direct binding of p63 and p53 correlates with the modification of gene expression in C2C12 muscular cells.

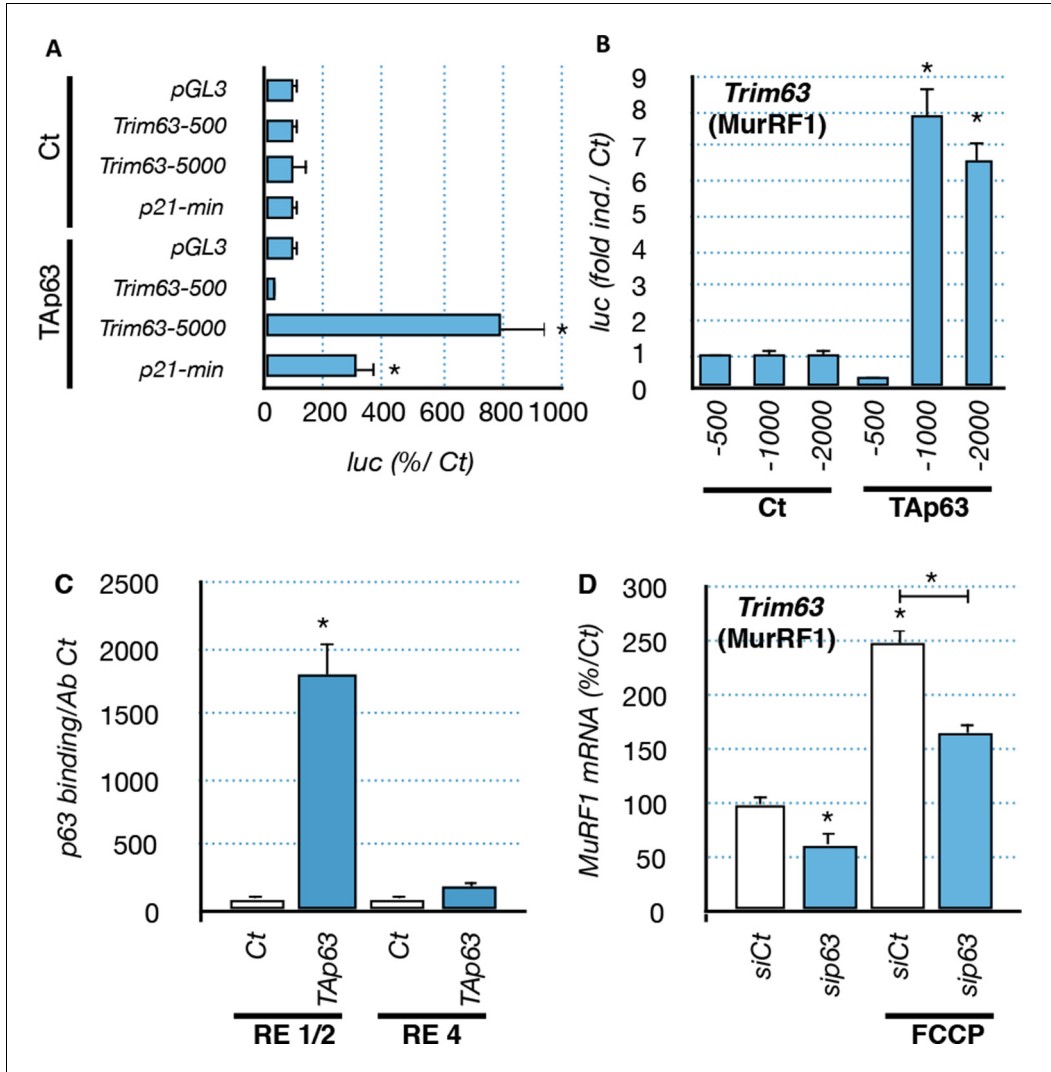

**Figure 7.** Regulation of *Trim63* promoter by p63. (A, B) *Trim63* promoter reporter constructs were co-transfected with pCDNA3 (Ct) or TAp63 into C2C12 cells and luciferase activity was assessed 16 hr later. pGL3 was used as a negative control. Bars correspond to means with SD (n = 3). *p<0.01 as calculated by a one-way ANOVA test followed by a Tukey post-test. (C) Chromatin immunoprecipitation assay was performed on the *Trim63* promoter using RT-qPCR on RE1/2 and RE4 (see *Figure 6A*). Bars correspond to means with SD (n = 3). *p<0.01 as calculated by a one-way ANOVA test followed by a Tukey post-test. (D) *Trim63* mRNA levels were assayed in C2C12 cells by RT-qPCR after TAp63 silencing by siRNA for 36 hr and after treatment with FCCP for 12 hr. Bars correspond to means with SD (n=3). *p<0.01 as calculated by a one-way ANOVA test followed by a Tukey post-test.

The following figure supplements are available for figure 7:

**Figure supplement 1.** Regulation of *Trim63* by p53 and p73 proteins.

**Figure supplement 2.** Impact of p63 on C2C12 cell survival.

## Discussion

In this study, we developed a comprehensive approach combining biopsies from ALS patients, transgenic animal model of ALS and myoblastic cell lines to analyse the expression and the possible function of *P63*, a member of the p53 family, in muscle atrophy.

# Regulation of p63 expression during muscular atrophy in ALS patients and in ALS murine models

Our results demonstrate that there is a complex p53-like response developed by the atrophic muscle during ALS progression. This assertion is first based on the bioinformatic signalling pathway analyses of 4 independent microarray experiments performed on muscle biopsies of ALS patients as well as two different mouse models of ALS. These analyses pointed out a deregulation of p53 as one of the only three transcription factors deregulated in all four experiments, and common between mouse and human patient samples. Moreover, detailed expression profile analyses of several p53 target genes (*Cdkn1a, Gadd45a, Pmaip1*) or the p53 family member, *P63*, showed that their expression correlated significantly with the severity of the pathology in humans. The signalling pathway analyses were confirmed with groups of individuals independent of those used for the microarrays and by additional experimental methods. RT-qPCR confirmed the induction of multiple target genes of the p53 family. In addition, expression analysis of p53 family members indicated that in ALS the *P63* gene seems more likely to play a regulatory role as the TAp63 isoforms are strongly upregulated and localized in the nuclei of the fibers during the ALS pathology (*Figures 1*, *3*, *4*, *5*, figure supplement 2, 3, 4, 7). Our observation that the deregulation of p63 and p63 target genes occurs in muscle of ALS patients that have not been selected for a particular genetic alteration indicates that these deregulations are likely to be a common feature in ALS, independently of whether it is SOD1 that is mutated or another gene. Additional experiments using other mouse models for TDP43 or FUS might confirm that. It was previously reported that p63 participates in muscle cell differentiation and metabolism, and contributes to cardiac muscle development (*Cam et al., 2006*; *Martin et al., 2011*; *Rouleau et al., 2011*; *Su et al., 2012*; *Osada et al., 1998*). Now, by combining biopsies from ALS patients and an animal model for ALS, the present study provides the first solid evidence that p63 might also participate in muscular atrophy.

Although our results indicate that TAp63 is strongly induced in muscle atrophy during ALS, we cannot exclude the possible contribution of p53 and p73 proteins due to the fact that their mRNA expression is upregulated, although to a much weaker extent than TAp63 (Figure 3—figure supplement 4). In addition, we detected p53 and p73 protein expression in muscles tissues. Several studies support this possibility by showing that p53 and p73 play a role in muscle cell differentiation, cachexia and survival (*Schwarzkopf et al., 2006*; *Cam et al., 2006*, *Soddu et al., 1996*; *Tamir and Bengal, 1998*; *Porrello et al., 2000*; *Weintraub et al., 1991*). However, genetic inactivation of p53 does not affect ALS progression, muscle development or muscle regenerative capacity (*Donehower et al., 1992*; *White et al., 2002*; *Kuntz et al., 2000*; *Prudlo et al., 2000*). Nevertheless, our results suggest that the absence of p53 could be compensated by p63 or even p73. The slight increased in p53 protein levels observed in protein extracts of muscle from SOD1(G86R) mice might be caused by the production of ROS that stabilized p53 through post-translational modifications as previously described in other stresses (*Vurusaner et al., 2012*). In addition, the slight increase in p53 RNA level we observed might also contribute.

The causes of p63 regulation during ALS muscle atrophy seemed complex and reflect the debated etiology of the pathology (*Yamanaka et al., 2008*; *Boillée et al., 2006*; *Wong and Martin, 2010*; *Chen et al., 2013*). For instance, we showed that *P63* deregulation could have an intercellular origin represented by the loss of interaction between the muscle and the nerve cells, as provoked in the nerve crush experiment (*Figure 5*). Hence, the degeneration of the motor neurons that is characteristic of ALS would be sufficient to explain the increased expression of TAp63 and its target genes in ALS. However, we also observed that *P63* deregulation can have an intrinsic origin resulting from expression of SOD1(G86R) in muscle cells causing intracellular stresses (*Figure 5*). Indeed, ALS-associated SOD1 mutations have been shown to induce SOD1 protein aggregates and mitochondrial dysfunction in muscle cells (*Pansarasa et al., 2014*). Interestingly, we observed that both activation of a protein aggregate stress response pathway or mitochondrial dysfunction could induce a TAp63 response. This finding is supported by a previous report showing that *tp63* is an effector of the ER stress pathway in zebrafish allowing the regulation of the pro-apoptotic gene *bbc3* (puma) (*Pyati et al., 2011*).

Protein aggregates, mitochondrial stress and oxidative stress triggered selective complex stress pathways named ER stress (or UPR, unfolded response) or mitochondrial stressed pathways that utilize several common transcription factors as effectors, such as ATF4, ATF6, CHOP and XBP1

(*Senft and Ronai, 2015*; *Broadley and Hartl, 2008*; *Lee, 2015*; *Michel et al., 2015*). Therefore, we investigated whether these effectors could drive the expression of p53 family protein. We showed that some of these transcription factors, notably ATF4 and ATF6, were able to induce the RNA levels of TA isoforms of *Trp63* and *P73* in C2C12 cells (*Figure 5—figure supplement 2*). However, bioinformatic analyses did not reveal potential canonical binding sites for these transcription factors neither in promoters of TA isoforms of *P63* nor *P73*, suggesting that the regulation might occur through indirect mechanisms that remain to be identified.

## Function of the p53 response in muscular atrophy during ALS

Based on the literature, the p53 family could mediate different cellular outcomes in muscles and therefore on muscle pathology. p53/p63/p73 proteins have been linked to cell death, differentiation, metabolism, ER stress induction and ROS defence, which have all been reported during ALS (*Hart, 2006*; *Barber and Shaw, 2010*; *Aguirre et al., 2005*; *Nishitoh et al., 2008*; *Manfredi and Xu, 2005*). Our study revealed that the majority of the p53/p63/p73 target genes upregulated during ALS in the atrophic muscles are connected to cell death (*Gadd45a, Peg3, Perp, Pmaip1, Bax, Siva, Eda2r, Wig1/Pag608*) (*Figure 1* and *Table 1*). Genes connected to other functions, such as ER stress (*chop, bip, xbp1, scotin*) or energy metabolism (*Tigar, Sesn1, Sesn2, Sco2*) seem to be less consistently regulated, as some are upregulated (*Sesn1, Sesn2, Xbp1*), while others are downregulated (*Sco2, Tigar, Chop, Bip*, see *Table 1*). Therefore, it seems more likely that p53 family members, notably TAp63, function in ALS is connected to muscular atrophy via control of muscle cell survival and catabolism. This hypothesis is further supported by three of our results.

First, the p53 family members TAp63, p53 and TAp73 induce the muscle atrophy effector gene *Trim63 (MuRF1)*, most likely via a direct binding to the *Trim63* promoter (*Figure 7*, *Figure 7—figure supplement 1*). Second, overexpression of TAp63 induces cell death in C2C12 myoblasts (*Figure 7—figure supplement 1*). Third, overexpression of △Np63 protects myoblastic cells against stresses (*Figure 7—figure supplement 2*). Although these results were obtained in a myoblastic cell line, they are consistent with numerous other studies describing the ability of p63 to control cell death in various pathophysiological conditions. To establish the exact pathophysiological importance of TAp63 upregulation in ALS represent a difficult challenge. Indeed, we already observed that in vitro the silencing of TAp63 with siRNA does not entirely abolish the expression of *Trim63* (*Figure 7*) and does not significantly reduce cell death induced by stresses (*Figure 7*, *Figure 7—figure supplement 2*). This has several reasons. The first is that the expression of *Trim63* involves several transcription factors, such as FOXO1 and the glucocorticoid receptors that certainly participate in the regulation of *Trim63* during ALS. Indeed, the coordinated silencing of p53, TAp63 and TAp73 did not completely abolish *Trim63* RNA levels (*Figure 7—figure supplement 1*), supporting the involvement of other types of transcription factors. The second reason is intrinsic to the p53 family. Indeed, we already know that p53, p63 and p73 have some redundant functions and target genes. We have also already established that p53 and p73 are expressed in muscles during the pathology (*Figure 4—figure supplements 1,2*) and also bind to the *Trim63* promoter (*Figure 7—figure supplement 1*). Therefore in absence of TAp63, p53 and/or TAp73 might replace it in some conditions. For example, we observed an upregulation of TAp73 in C2C12 cells when TAp63 is silenced. This compensatory mechanism might therefore also explain why TAp63 siRNA do not protect C2C12 cells from death, in contrast to the expression of the △Np63 isoform that could inhibit p53, TAp73 and TAp63 function altogether (*Figure 7—figure supplement 2*).

ALS patients are currently diagnosed at a stage where denervation and muscular alterations already are established and, because of the lack of curative treatment, lead to death within 2 to 5 years. The results presented here suggest that p53 family members, via the regulation of selected target genes such as *Eda2r, Peg3* but also, as we show, via *Trim63*, might contribute to muscle catabolism in these patients. It remains to be established whether this signalling pathway is uniquely critical for muscular atrophy during ALS or whether it is common to other muscular atrophies occurring in pathologies such as cachexia, diabetes, and others.

## Materials and methods

### SOD1-G86R mice

SOD1(G86R) mice were genotyped as described in (*Ripps et al., 1995*). For surgery, 80-day-old FVB mice were anesthetized and both sciatic nerves were exposed at mid thigh level and crushing was performed (or not – CT) with a forceps during 20 s ~5 mm proximal to the trifurcation. Control animals used in the experiments were wild-type littermates. Randomization was performed based on body weight. Time course for animal pathology was performed based on a previous study on denervation and muscle atrophy (*von Grabowiecki et al., 2015*). Animal experiments were performed following the European guidelines and protocols validated by the local ethical committee.

### Cell culture

C2C12 cells were obtained from ATCC (ATCC CRL-1772) and grown in DMEM (Dulbecco's modified Eagle's medium; Life Technology, Carlsbad, CA) with 10% fetal bovine serum (Life Technology) at 37°C in a humidified atmosphere and 5% $CO_2$. *Mycoplasma* contamination has been tested negatively using PlasmoTest (Invivogene, San Diego, CA). Differentiation of C2C12 cells was performed using 2% horse serum at 90% cell confluence.

### Quantitative PCR

TRIzol (Invitrogen, Carlsbad, CA) was used to extract RNA. One µg of RNA was used for reverse transcription (iScript cDNA kit, Bio-Rad, France) and qPCR was carried out (iQ SYBR Green, Bio-Rad) (*Supplementary file 1*). Expression levels were normalized using either 18S, TBP or RPB1 as previously described (*Vidimar et al., 2012*).

### Western blotting

Cells or tissue were lysed with LB (125 mM Tris-HCl pH 6.7, NaCl 150 mM, NP40 0.5%, 10% glycerol). Proteins were denatured and deposited directly (75 µg of proteins) onto a SDS-PAGE gel, or they were precipitated (2 mg of proteins) with a p63 antibody and G Sepharose beads before separation. Western blotting was performed using antibodies raised against p53 (rabbit anti-p53, FL-393, Santa Cruz Biotechnology, Dallas, TX), p63 (mouse anti-p63, 4A4, Santa Cruz Biotechnology; p63, Abcam, France) or TAp63 (Biolegend, CA). Secondary antibodies (anti-rabbit, anti-mouse: Sigma, France) were incubated at 1:1000. Loading was controlled with actin (rabbit anti-β-actin, Sigma, 1:4000) or TBP (anti-TBP 1:1000, Santa Cruz Biotechnology) (*Antoine et al., 1996*).

### Transfection and luciferase assays

Cells were transfected by polyethylenimmine (PEI)-based or JetPrim (Polyplus, Strasbourg, France) as previously described (*Gaiddon et al., 1999*). For luciferase assays, cells were seeded in 24-well plates, and transfected with the indicated expression vectors (200 ng) and reporter constructs (250 ng) (*Sohm et al., 1999*). Luciferase activity was measured in each well 24 hr later and results were normalized with a CMV-driven reporter gene (*Benosman et al., 2011*). The -1584 ΔNp63 luc and -46 ΔNp63 luc constructs were previously described (*Romano et al., 2006*). The *Trim63* luc constructs were previously described (*Waddell et al., 2008*). SiRNA tranfection was performed using 30 nM of siRNA and with RNAiMAX protocol as described by the provider (Life Technology). TAp63 siRNA sequences were covering the sequence: GAA CUU UGU GGA UGA ACC UCC GAA.

### Chromatin immunoprecipitation (ChIP) assay

ChIP assays were performed using the standard protocol from the Magna ChIP G kit (Millipore). C2C12 lysates were sonicated 12 times at 10% power. For each 1 million cells, 1 µg of antibody was used. p63 was immunoprecipitated with a mouse antibody raised against total p63 (4A4, Santa Cruz Biotechnology). Mouse-anti-RAB11A was used as negative control (Santa Cruz Biotechnology).

### Microarrays analyses

ECL files from microarray experiments (E-MXP-3260; E-GEOD-41414; E-TABM-195; E-GEOD-16361) were obtained form the Array Express database (EMBL-EBI). Each experiment was first analyzed individually using AltAnalysis software (*Emig et al., 2010*). Deregulated gene were identified based on

two fold change expression and t-test p-value <0.05. Deregulated genes were then analyzed by GO-Elite with Prune Ontology term using Z-score (cutoff 1.96, p-value 0.05) and Fisher's Exact Test for ORA (2000 permutations) for over-representation in selected biological processes in several resources: Gene Ontology, MPhenoOntology, Disease Ontology, GOSlim, PathwayCommons, KEGG, Transcription Factor Targets, miRNA Targets, Domains, BioMarkers, RVista Transcription Sites, DrugBank, BioGrid.

## Immunohistochemistry

Mouse gastrocnemius muscles were sampled, submersed in freezing medium (Tissue-Tek O.C.T compound, Sakura, Japan) and immediately frozen in a nitrogen-cooled isopentane bath. Muscles were sliced in transversal axis at 14 µm in a cryostat (Leica CM3050S, Leica, France) and placed on slides covered with 0.5% gelatine. The samples were then dried for 20 min on a hot plate and fixed in 4% paraformaldehyde for 10 min . After a 5 min wash with PBS, the samples were permeabilized with 3% Triton X-100 in PBS for 10 min, washed with TBS, incubated in 100 mM glycine in TBS for 20 min and finally washed again in PBS. The samples were incubated with mouse antibody raised against p63 (p63 clone 4A4, Santa Cruz Biotechnology) at 1:100 with 0.1% Triton X-100 in PBS (Triton buffer) overnight at room temperature. They were then washed three times with Triton buffer for 10 min and incubated with cyanine 3-coupled goat anti-mouse antibody (Jackson ImmunoResearch, West Grove, PA) at 1:1000, as well as with 1 µg/ml Hoechst 33,342 (Sigma, France), in Triton buffer at room temperature for 1 hr. After washing three times with Triton buffer, the slides were covered with mounting medium (Aqua-Poly/Mount, Polysciences, Warrington, PA) on glass slips and observed by confocal microscopy (Zeiss, Germany). Antibody specificity was verified with slides probed with only the secondary antibody.

## Acknowledgement

We thank Dr. Sinhna S (Buffalo, USA) for providing the △Np63 promoter constructs and Pr. Bodine for the *Trim63* reporter genes. We are also grateful to the Neuromuscular Unit (BioBank of Skeletal Muscle, Nerve Tissue, DNA and cell lines, Milan, Italy) for the generous gift of ALS biopsy samples that were obtained with the consent of patients and under the ethical rules established by the Italian ethical counsel. This project is supported by the CNRS (France) (CG), FNR (Luxembourg) (YvG), ARC, Ligue contre le Cancer, AFM, COST CM105. We are also thankful for the technical support of A Picchinenna and E Martin.

## Additional information

### Funding

| Funder | Author |
| --- | --- |
| Fonds National de la Recherche Luxembourg | Yannick Von Grabowiecki |
| Institut National de la Santé et de la Recherche Médicale | Véronique Devignot<br>Isabelle Gross<br>Georg Mellitzer<br>Christian Gaiddon |
| Association Française contre les Myopathies | Christian Gaiddon |
| Association pour la Recherche sur le Cancer | Christian Gaiddon |
| Centre National de la Recherche Scientifique | Christian Gaiddon |

The funders had no role in study design, data collection and interpretation, or the decision to submit the work for publication.

## Author contributions
YvG, CG, Conception and design, Acquisition of data, Analysis and interpretation of data, Drafting or revising the article; PA, SB, VD, Conception and design, Acquisition of data, Analysis and interpretation of data; OB, SM, Acquisition of data, Analysis and interpretation of data; LP, Analysis and interpretation of data, Drafting or revising the article, Contributed unpublished essential data or reagents; IG, Did the ChIP experiments added in the revised version of the manuscript (design, acquisition of data, analysis) and participated in revising the article, Conception and design, Acquisition of data, Analysis and interpretation of data, Drafting or revising the article; GM, Analysis and interpretation of data, Drafting or revising the article; JLGdA, Acquisition of data, Analysis and interpretation of data, Drafting or revising the article, Contributed unpublished essential data or reagents

## Author ORCIDs
Yannick von Grabowiecki, http://orcid.org/0000-0003-2189-6953
Christian Gaiddon, http://orcid.org/0000-0003-4315-3851

## Ethics
Animal experimentation: This study was authorised and performed in strict accordance with the recommendations in the Guide for the Care and Use of Laboratory Animals of the Local Ethical committee (Permit Number: 67/34577). All surgery was performed under sodium pentobarbital anesthesia, and every effort was made to minimize suffering.

## Additional files

### Supplementary files
• Supplementary file 1.
• Supplementary file 2.

### Major datasets
The following previously published datasets were used:

| Author(s) | Year | Dataset title | Dataset URL | Database, license, and accessibility information |
|---|---|---|---|---|
| Demougin Philippe | 2014 | Muscle Gene Expression Is a Marker of Amyotrophic Lateral Sclerosis Severity | https://www.ebi.ac.uk/arrayexpress/experiments/E-MEXP-3260/?query=ALS+muscle | E-MEXP-3260 |
| Bernardini C, Ricci E | 2014 | Gene expression analysis of ALS skeletal muscle | https://www.ebi.ac.uk/arrayexpress/experiments/E-GEOD-41414/?query=ALS+muscle | E-GEOD-41414 |
| Demougin Philippe | 2008 | Transcription profiling of skeletal muscle from amyotrophic lateral sclerosis sod1(G86R) axotomized mice and control mice to monitor denervation-dependent gene expression in an Amyotrophic lateral sclerosis (ALS) mouse model | https://www.ebi.ac.uk/arrayexpress/experiments/E-TABM-195/?query=ALS+muscle | E-TABM-195 |
| Ratti A, Volta M, Calza S | 2014 | Gene expression profiling of muscles from transgenic humanSODG93A mice at symptomatic stage | https://www.ebi.ac.uk/arrayexpress/experiments/E-GEOD-16361/?query=ALS+muscle | E-GEOD-16361 |

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
