## [Decision Letter]

Thank you for submitting your work entitled "TAp63 is upregulated in muscular atrophy during ALS and induces the pro-atrophic ubiquitin ligase *murf1*" for consideration by *eLife*. Your article has been reviewed by two peer reviewers, one of whom is a member of our Board of Reviewing Editors, and the evaluation has been overseen by James Manley as the Senior Editor.

The reviewers have discussed the reviews with one another and the Reviewing editor has drafted this decision to help you prepare a revised submission.

Summary of the findings in this paper:

The authors have used bioinformatics approaches to identify the p53 family, TAp63 in particular, as a player in muscle atrophy in diseases such as ALS. Using gene expression meta-analyses in ALS patients they discovered that several p53 family target genes were strongly upregulated and the extent of their upregulation correlated with severity of the disease as measured by degree of muscle injury. Similar results were seen in muscles of SOD1 (G86R) mice. Of the different p53 family that were assayed there was a strong increase specifically in TAp63 and the authors conclude that of the p53 family, it is this isoform that is responsible for the upregulation of these genes. They test this by performing sciatic nerve crush experiments and show that this causes a huge increase in TAp63 and expression of p21 and GADD45a target genes. In fact they also find that levels of the p63 delta N isoform (that in some settings counteracts the TA isoform) actually decrease in this experimental setting. They then identify *murf1*, a gene whose product has been shown to be involved in muscle atrophy as a p63 target gene and provide evidence that of the various p53 family member isoforms, TAp63 is the strongest inducer of this gene in C2C12 myoblasts, cells that were used so that transfection experiments could be performed in a relevant muscle cell line. They also provide evidence that upon siRNA mediated knock-down of p63, there is a significant reduction in expression of *murf*1 in these cells.

Summary of referees' comments:

The authors have provided solid evidence that the p53 family genes along with their targets are transcriptionally regulated by ALS-associated SOD mutations, that is, SOD1 protein aggregate stress response pathway or mitochondrial dysfunction. Since the p53 family proteins and their major targets are involved in apoptosis and other related cellular response, it is likely that one of the mechanisms by which muscle atrophy is induced by abnormal apoptotic and other related responses. Overall this is a very interesting paper that links for the first time p63 and muscle degeneration and ALS. Although there are some gaps due to difficulties in obtaining muscle cell lines from patients, their data make a quite convincing case for the TA isoform of p63 being a factor in regulating muscle deterioration via up-regulation of *murf1*. This is a novel set of observations and the study is recommended to the journal with the condition that the authors can address most if not all the comments below:

Essential revisions:

1) The authors showed that knockdown of p63 reduces expression of *murf1* in C2C12 cells (Figure 7) but does not improve survival of these cells after treatment with FCCP or menadione in Figure 7—figure supplement 2). It is possible that p63 and p73 may cooperate in regulation of *murf1*. Although TAp63 seems to be the dominant p53 family member isoform in regulating *murf1*, expression of TAp73 was also able to do so, albeit less efficiently. It would be informative if the authors could determine whether (a) endogenously expressed p73 can bind to and regulate *murf1* in C2C12 cells and (b) whether its knockdown affects *murf1* expression or (c) whether, when combined with p63 knockdown, there was a bigger impact on *murf1* than either alone. This might explain why there is not a big effect on cell death of p63 knockdown in these cells in response to stresses.

2) Under most of the stress conditions, p53 expression is upregulated due to post-translational modifications. Here, the authors showed that the induction of p53 is due to increased mRNA expression (possibly transcription?). Is this due to the type of stresses or anything else? Can they detect p53 protein? While the level of p53 is low in most of normal tissues and cells, many studies have shown that p53 can be detected in normal mouse tissues and cells. The authors may have to repeat the experiments, especially with anti-mouse 1C12 antibody.

3) The authors should address with data if possible or at least in the Discussion what transcription factor(s) are induced by SOD1 protein aggregate stress response pathway or mitochondrial dysfunction, which then activate the p53 family proteins via transcription.

---

## [Author Response]

Essential revisions: 1) The authors showed that knockdown of p63 reduces expression of murf1 in C2C12 cells (Figure 7) but does not improve survival of these cells after treatment with FCCP or menadione in Figure 7—figure supplement 2). It is possible that p63 and p73 may cooperate in regulation of murf1. Although TAp63 seems to be the dominant p53 family member isoform in regulating murf1, expression of TAp73 was also able to do so, albeit less efficiently. It would be informative if the authors could determine whether (a) endogenously expressed p73 can bind to and regulate murf1 in C2C12 cells and (b) whether its knockdown affects murf1 expression or (c) whether, when combined with p63 knockdown, there was a bigger impact on murf1 than either alone. This might explain why there is not a big effect on cell death of p63 knockdown in these cells in response to stresses.

As suggested, we have performed both p73 silencing and p73 ChiP experiments. TAp73 proteins were able to bind the *murf1* promoter. However, the silencing of TAp73 did not significantly affect the RNA levels of *murf1,* suggesting a compensatory mechanism. As we also detected a slight increase in p53 protein levels in the muscle of SOD1(G86R) animals, we additionally investigated the binding of p53 on *murf1* promoter and the impact of p53 silencing on *murf1* RNA levels. P53 was able to bind on *murf1* promoter and its silencing significantly reduced *murf1* RNA level. However, even the combined silencing of TAp63, TAp73 and p53 did not completely abolished *murf1* RNA level. This new set of experiments was added to the manuscript and confirmed the transcriptional regulation of *murf1* by p53 family members as well as the implication of other mechanisms.

*2) Under most of the stress conditions, p53 expression is upregulated due to post-translational modifications. Here, the authors showed that the induction of p53 is due to increased mRNA expression (possibly transcription?). Is this due to the type of stresses or anything else? Can they detect p53 protein? While the level of p53 is low in most of normal tissues and cells, many studies have shown that p53 can be detected in normal mouse tissues and cells. The authors may have to repeat the experiments, especially with anti-mouse 1C12 antibody.*

We thank the referees for suggesting the use of the 1C12 p53 antibody. Indeed, using this antibody we succeeded in following p53 protein levels in protein extracts of muscles from WT and SOD1(G86R) mice, in contrast to our previous attempts with other antibodies. The new data show a slight increase in p53 protein levels, which encouraged us to further investigate the possible regulation of *murf1* by p53 (see above). Unfortunately and despite multiple attempts we did not succeed in showing any specific staining in muscle tissues using this antibody. The company (Cell Signaling) providing this antibody confirmed that it is not useful for immunohistochemistry with mouse tissues. It remains to establish the mechanism involved in the upregulation of p53 proteins levels. One hypothesis might be that the oxidative stress described in tissues during ALS might trigger canonical post-translational modifications of p53 and its stabilization as previously described for cisplatin for instance. In addition, the slight increase in p53 RNA levels might contribute to the induction of p53 protein levels. These points have been added to the Discussion.

*3) The authors should address with data if possible or at least in the Discussion what transcription factor(s) are induced by SOD1 protein aggregate stress response pathway or mitochondrial dysfunction, which then activate the p53 family proteins via* transcription.

As suggested we have investigated and discussed the possible involvement of several transcription factors that might activate the transcription of p53 family proteins and that are connected to ER stress or mitochondrial stress. More precisely, we focused on ATF4, ATF6, CHOP and XBP1. These 4 transcription factors have been linked to ER stress and mitochondrial dysfunction. We have now added data showing that at least ATF4 and ATF6 can contribute to the induction of TAp63 and TAp73 RNA level (Figure 5—figure supplement 2). However, bioinformatics did not allow us to locate any canonical binding sites for these transcription factors in TAp63 or TAp73 promoters, suggesting that the regulation might be indirect. We have added these new data in the manuscript and further discussed them in the Discussion.